# Phenological strategies of an evergreen tree in the Caatinga

**Fernanda Moura Fonseca Lucas**[1], **Kyvia Pontes Teixeira das Chagas**[2],
**Ageu da Silva Monteiro Freire**[2], **Vivian Raquel Bezerra de Sousa**[2], **Fábio de Almeida Vieira** [ID][2]*

1 Federal University of Espírito Santo - UFES, Department of Forestry and Wood Sciences, Avenue Governador Lindemberg, Jeronimo Monteiro, Brazil, 2 Federal University of Rio Grande do Norte - UFRN, Academic Unit Specialized in Agricultural Sciences, Macaíba, Brazil

* fabio.almeida@ufrn.br

## Abstract

Seasonally Dry Tropical Forests (SDTF) account for 40% of global tropical forests, with the Caatinga standing out as the largest continuous formation of this type. However, the region faces severe threats, such as deforestation and desertification, which require urgent conservation efforts. In this context, understanding the adaptive strategies of native species becomes essential to support management actions. This study aims to identify the phenological strategies of *Sarcomphalus joazeiro* (Mart.), a species of high ecological, cultural, and economic importance in the region. Over two years, intrapopulation monitoring of vegetative and reproductive phenophases was conducted in a forest fragment in Rio Grande do Norte, Brazil, evaluating phenophase seasonality, reproductive synchrony, correlation with meteorological variables (precipitation, relative humidity, and air temperature), and fruit and seed biometrics. The results revealed that the flowering and fruiting of *S. joazeiro* are annual, synchronized, and occur during the dry season, highlighting an adaptive reproductive strategy and providing an important food source for the fauna. The species exhibited a weak correlation between its phenophases and meteorological variables, emphasizing its resistance to adverse climatic conditions. These characteristics make *S. joazeiro* unique among SDTF trees and underscore its ecological importance and potential for management and degraded area restoration strategies. Phenological studies with other Caatinga species are recommended to deepen understanding of biota-climate interactions and to contribute to effective conservation strategies.

## Introduction

Seasonally Dry Tropical Forests (SDTF) encompass a type of vegetation within a complex of typologies, considered a global metacommunity [1]. SDTFs cover approximately 40% of the world's tropical forests and are primarily located in the Neotropical region [2]. In Brazil, they are predominantly concentrated in the northeastern region, within the biome known nationally as the Caatinga.

The Caatinga is the largest continuous formation of SDTF in the world [3] characterized by high diversity and composed of a flora with over 4,000 species, approximately 20% of which are endemic [4,5]. Its remarkable diversity is evidenced by a species/area ratio nearly twice

**Data availability statement:** All relevant data are within the manuscript and its Supporting Information files.

**Funding:** This work was financed in part by the Office to Coordinate Improvement of Higher Education Personnel - Brazil (CAPES) - Finance Code 001. The authors acknowledge the Conselho Nacional de Desenvolvimento Científico e Tecnológico (CNPq) for their financial support (Grant no. 407700/2023-4). The funders had no role in study design, data collection and analysis, decision to publish, or preparation of the manuscript.

**Competing interests:** NO authors have competing interests.

as high as that of the Amazon Rainforest [6]. Additionally, plant communities adapted to drought have resulted from a mosaic of evolutionary histories involving accumulated immigration from adjacent biogeographic regions and recent in situ speciation events [6].

*Sarcomphalus joazeiro* (Mart.), commonly known as juazeiro, is an evergreen species endemic to northeastern Brazil, belonging to the Rhamnaceae family. It occurs in the Caatinga biome and in transition zones. Widely used, it is notable for its application in traditional medicine [7,8], animal feed during droughts, cosmetics production, pharmacology, forest restoration, and as a source of thermal comfort due to its large canopy [9]. Its extracts exhibit antifungal, antibacterial, antiparasitic [10,11], antioxidant, and whitening properties, being used in dental treatments [12,13] and in the formulation of products like shampoos and detergents due to the presence of bioactive saponins with anti-helminthic effects [14]. Beyond its economic and medicinal relevance, the species is essential for the ecosystem, providing pollen and nectar to native bees of the Meliponini tribe [9], though studies on its ecology and conservation are still scarce [15].

Despite the importance of such species, the Caatinga faces increasing threats due to the exposure of its forest remnants to anthropogenic pressures such as agricultural expansion, deforestation, accelerated desertification, and scientific neglect compared to other tropical forests [16–18]. In this context, even housing more than 27 million people, the Caatinga remains one of the least studied global ecological regions, highlighting the urgent need for conservation and ongoing research to understand and protect its physiognomies [19].

Understanding plant phenology is a fundamental aspect of ecology, providing insights into community dynamics and temporal fluctuations in resources available to consumers [20]. Essentially, phenological study refers to the investigation of vegetative and reproductive development patterns throughout the year [21]. In recent decades, this type of study has provided information about ecosystem responses to climate change, contributing to making phenology a predictive science [22]. However, despite their importance, few studies provide ecological information on the phenology of forest species, especially in the Caatinga. Some more recent studies have carried out phenological analyses using satellite images, with the aim of optimizing field monitoring [23]. Along with phenology, fruit and seed biometrics play an important role in restoration projects and systematic studies, helping to identify genetic variability within species populations and their interaction with environmental factors [24].

Different ecological strategies have been reported in seasonal tropical forests [25,26]. However, given the uniqueness of the Caatinga and the lack of scientific contributions regarding its endemic species, understanding processes such as growth and reproduction can support the planning and actions of forest management, enabling a connection between resource exploitation and conservation.

In light of the above, this research was conducted to observe the phenological and biometric traits of *Sarcomphalus joazeiro* (Mart.), aiming to identify possible strategies adopted by the species to ensure its survival and reproduction under drought conditions, as well as to recognize development patterns that can support sustainable exploitation and management of the species. To investigate the ecological strategies adopted by the species, the following research questions were established: i) Is there a strong correlation between reproductive phenophases and meteorological variables? ii) Does the species exhibit intrapopulational synchrony and an aggregated distribution pattern for reproductive events?

## Materials and methods

### Study species

*Sarcomphalus joazeiro* Mart. is widely distributed in the Caatinga and has multiple uses, standing out for its ornamental qualities and economic and cultural importance in forestry,

food and medicine, especially for traditional communities in the Brazilian northeast region [27,28]. With individuals reaching up to 16 meters in height and displaying a wide crown, this species has deep roots, branches with thorns (typical of xerophytic vegetation), three-nerved leaves, globose to ellipsoid drupes and greenish-yellow flowers arranged in terminal or auxiliary inflorescences [29,30].

## Study area

The research was conducted in a fragment of Seasonal Deciduous Forest, as classified by the Brazilian Institute of Geography and Statistics [31], named Mata do Olho d'Água, located in the municipality of Macaíba, Northeast region of Brazil, between the coordinates 5°53'07.1" S and 35°22'02.6" W, with an average elevation of 40 meters above sea level and an area of approximately 270 hectares. The local forest represents a transition area between the Atlantic Forest and the Caatinga, both floristically and environmentally [32]. The climate is characterized as a transition between the As' and BSh' types according to the Köppen classification [33], with an average temperature of 25.8 °C and an annual average rainfall of 1,134 mm, marked by high temperatures throughout the year with rainfall in autumn and winter (Fig 1). The study area is part of the campus of the Federal University of Rio Grande do Norte, and no authorization was required for access.

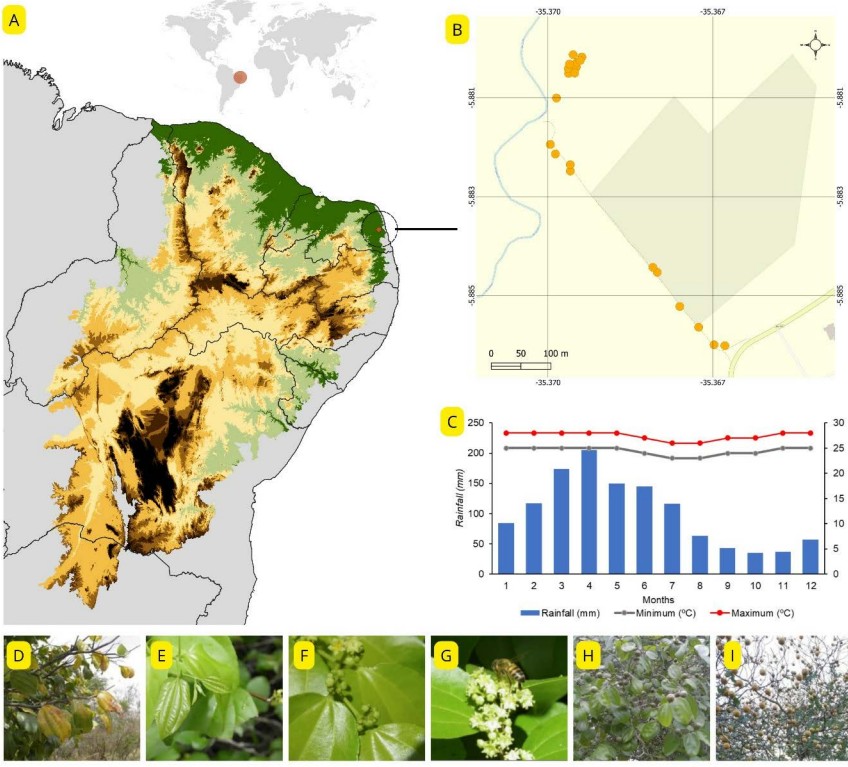

**Fig 1. (A) Distribution of the Caatinga biome among Brazilian states; (B)** *Sarcomphalus joazeiro* **population used in the study, with map data from OpenStreetMaps (accessed on 2024-12-05, openstreetmap.org/copyright); (C) historical monthly averages of precipitation (bars) and minimum and maximum temperatures for the municipality of Macaíba – RN; (D-I) phenophases observed in the study: defoliation (D), leaf budding (E), flower buds (F), flowers anthesis (G), immature fruits (H), and ripe fruits (I).** The images (D-I), provided by the authors, are suitable for publication under the Creative Commons Attribution License (CC BY 4.0).

## Phenological data collection

For the phenological study, twenty individuals were sampled, constituting a population of *S. joazeiro*. The selection criteria included the presence of reproductive events, which indicated the trees were adults, and good phytosanitary conditions, characterized by the absence of visible diseases or parasitic infestations. The population had an average trunk height of 3.8 meters, total height of 5.8 meters, and a diameter at breast height (DBH) of 17.4 cm. Sampling was conducted using the trail method [34], with individuals distributed along a pre-existing access path. To account for local heterogeneity, half of the individuals were located at the forest edge, while the other half were in a moderately dense vegetation zone under the influence of competition. Each tree was permanently marked for subsequent monitoring.

A total of 54 observations were carried out every two weeks between June 2016 and July 2018, with the monthly average of observations being calculated at the end of each month. The vegetative phenophases of leaf budding and defoliation were evaluated, along with the reproductive phenophases of the presence of floral buds, flowers anthesis, immature fruits, and ripe fruits.

The assessment method employed was the Fournier index, a semi-quantitative method that determines the intensity at which the phenophase occurs. This method evaluates phenophases on a constant interval scale with five occurrence intensities ranging from 0 to 4. A value of zero represents the absence of the phenophase occurrence; 1 represents the occurrence of the phenophase with an intensity of 1 to 25% canopy coverage; value 2 represents the occurrence with an intensity of 26 to 50%; value 3 represents the occurrence within an interval of 51 to 75%; and value 4 represents the occurrence with 76 to 100% canopy coverage [35].

For each conducted observation, the sum of intensity values for all individuals was calculated separately for each phenophase. Subsequently, it was divided by the total number of evaluated individuals (N = 20). This result was then multiplied by 100 to transform it into a percentage (F% = ($\sum$ intensities/4N) * 100) [34].

## Seasonality of phenophases

To identify the seasonality of the species, circular statistical analyses were conducted using the Oriana® demo program, version 4.02 (available at https://www.kovcomp.co.uk/oriana/index.html). For this, the average date for each month was converted into angles based on the Julian calendar. The Rayleigh test was performed for each phenological stage, and the mean vector length (r) was calculated. This test evaluates the uniformity of data distribution throughout the year. If the Rayleigh test indicated non-uniformity ($p < 0.01$), the value of r for each phenophase was interpreted as a measure of seasonal intensity. The r vector represents the degree of aggregation near the mean angle, ranging from 0 to 1, where higher values reflect greater seasonality [36]. Additionally, the angle representing the mean date of phenophase intensity (mean angle – μ) was converted. The Watson-Williams test was also employed to compare results between the two assessment years ($p < 0.01$).

## Spatial distribution of reproductive synchrony

The spatial distribution of flower and fruit events was analyzed based on data from a reproductive date, in other words, an observation that recorded high occurrence and intensity of phenophases in the population. For this, the coordinates of all trees that exhibited flowering or fruiting events on that specific date were collected. The neighborhood density with the same intensity of reproductive events was estimated using Moran's I index with the assistance of the ROOKCASE program [37], along with exact Monte Carlo tests to assess the significance of autocorrelation values. A correlogram was constructed to illustrate autocorrelation for

different distance classes of mapped reproductive phenophases, enabling the evaluation of the proximity effect between sets of sample pairs.

## Correlation with meteorological variables

The relationships between the occurrence of each phenophase and the air temperature, precipitation, and relative humidity variables were tested using Spearman's correlation. The correlation was performed with climatic data from the first ($r_{S1}$), second ($r_{S2}$), third ($r_{S3}$), and fourth ($r_{S4}$) fortnights preceding the observation of phenological events. For the statistical analysis of the data, deviations from the normality of phenological data were assessed using the Lilliefors test to justify the non-parametric Spearman correlation. Meteorological data were obtained from the climatological automatic station of the National Institute of Meteorology (INMET) closest to the study area, located in the municipality of Natal, approximately 20 km away.

## Biometry of fruits and seeds

For biometric characterization, 200 ripe fruits were randomly collected from 5 trees in the population that were not monitored by phenology. The evaluated characteristics included fresh weight (g), length (mm), diameter (mm), and thickness (mm). After measurement, the fruits were depulped, and the same characterization was conducted for the seeds. Dimensions were determined using a digital caliper (precision of 2 mm), and fresh weight was determined using a precision analytical balance. Descriptive statistics were performed for the obtained data.

## Results

### Phenology and seasonality

By observing the phenophases of *S. joazeiro* over 2 consecutive years, important results were obtained that can help in defining management strategies for the species. the Watson-Williams test showed a significant difference in the distribution of phenological events between the two evaluation years; however, the mean date of occurrence was preserved for most phenophases (Table 1).

**Table 1. Statistical description of phenology and occurrence of seasonality for the *Sarcomphalus joazeiro* Mart. population.** Average angle (μ), number of observations of phenophase manifestation (N), Rayleigh tests (Z) to verify seasonality (p < 0.01), intensity vector r, and Watson-William's test between the years of observation (p < 0.01). $Year_1$ = 2016/2017 and $Year_2$ = 2017/2018.

| Phenophase | Year | μ | Mean date | N | Z | r | Watson-Williams test |
|---|---|---|---|---|---|---|---|
| Defoliation | $Year_1$ | 211.192° | August | 383 | < 0.001 | 0.158 | < 0.001 |
| | $Year_2$ | | | | ns[*] | – | |
| Leaf budding | $Year_1$ | 310.107° | November | 137 | < 0.001 | 0.408 | < 0.001 |
| | $Year_2$ | 342.591° | December | 316 | < 0.001 | 0.248 | |
| Flower buds | $Year_1$ | 330.685° | December | 68 | < 0.001 | 0.856 | < 0.001 |
| | $Year_2$ | 352.942° | December | 166 | < 0.001 | 0.845 | |
| Flowers anthesis | $Year_1$ | 340.305° | December | 96 | < 0.001 | 0.849 | < 0.001 |
| | $Year_2$ | 356.921° | December | 167 | < 0.001 | 0.827 | |
| Green fruits | $Year_1$ | 43.916° | February | 224 | < 0.001 | 0.778 | < 0.001 |
| | $Year_2$ | 73.355° | March | 284 | < 0.001 | 0.622 | |
| Ripe fruits | $Year_1$ | | | | | | ns[*] |
| | $Year_2$ | | | | | | |

*ns = not significant (p > 0,01).

The defoliation of *S. joazeiro* occurred throughout the study period, albeit with reduced intensity in most months (below 50%) (Fig 2). The species exhibited low or no seasonality between the evaluated years (Table 1). The month with the highest intensity was October, with the observed mean date in August. On the other hand, leaf flush showed seasonality in both years, albeit low r values ($r_1 = 0.408$, $r_2 = 0.248$). Peaks of new leaf production mainly occurred during the dry period, with mean dates occurring between November and December.

All reproductive events of *S. joazeiro* exhibit seasonality. Floral buds and flower production occurred simultaneously, beginning in the dry season, with December as the mean date (Fig 3). Additionally, the studied population shows high synchrony in reproductive events, as the activities were recorded simultaneously in 80% of the individuals.

Fruit production also occurred seasonally, with immature fruits being intensively produced starting in December, and the mean observation dates occurring in February and March (Fig 4). Maturation followed soon after, with the mean date in April, representing the most suitable period for collection. During the second year, the occurrence of ripe fruits extended frequently, albeit at low intensity, throughout almost all months.

## Spatial distribution of reproductive synchrony

According to the Moran's I index value (Fig 5), plants with higher flowering intensity exhibited a spatially aggregated pattern up to 36 meters (Moran's I = 0.290, $p = 0.028$, dashed line), as did fruiting (Moran's I = 0.527, $p = 0.026$, solid line). Flowering intensity increased between the classes of 92 to 104 meters but not observed for fruiting.

## Correlation with meteorological variables

*Sarcomphalus joazeiro* exhibited weak or no correlation with meteorological variables, indicating that air temperature, relative humidity, and precipitation had little influence on phenophases during the evaluated period (Table 2).

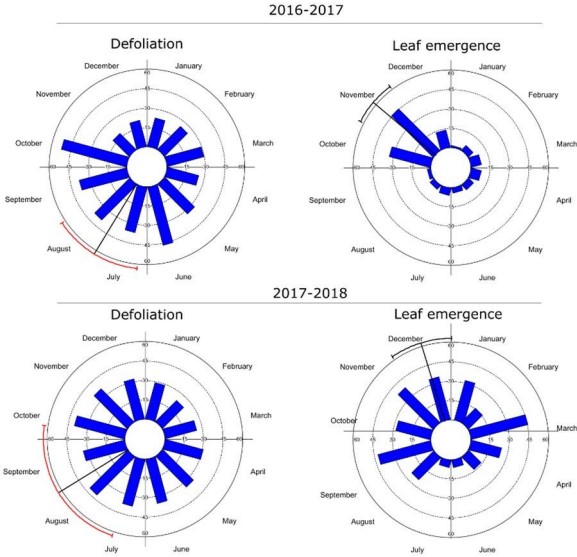

**Fig 2. Histograms of the intensity of vegetative phenophases (leaf budding and defoliation) of *Sarcomphalus joazeiro* Mart. over the months.** The arrow points to the average angle (or mean date) of the manifestation of the phenological event.

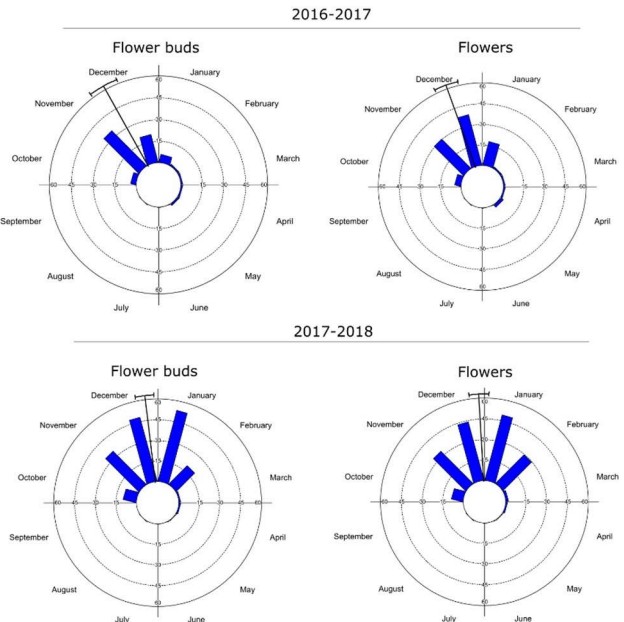

**Fig 3. Histograms of intensity portraying the flowering (floral buds and flowers) of** *Sarcomphalus joazeiro* **Mart. across the months.** The arrow indicates the average angle (or mean date) of the manifestation of the phenological event.

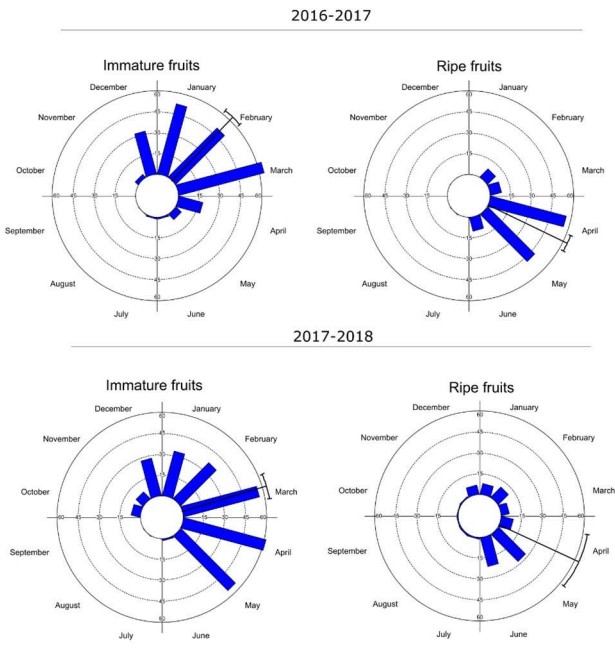

**Fig 4. Histograms of the intensity of fruiting (immature and ripe fruits) of** *Sarcomphalus joazeiro* **over the months.** The arrow points to the average angle (or mean date) of the manifestation of the phenological event.

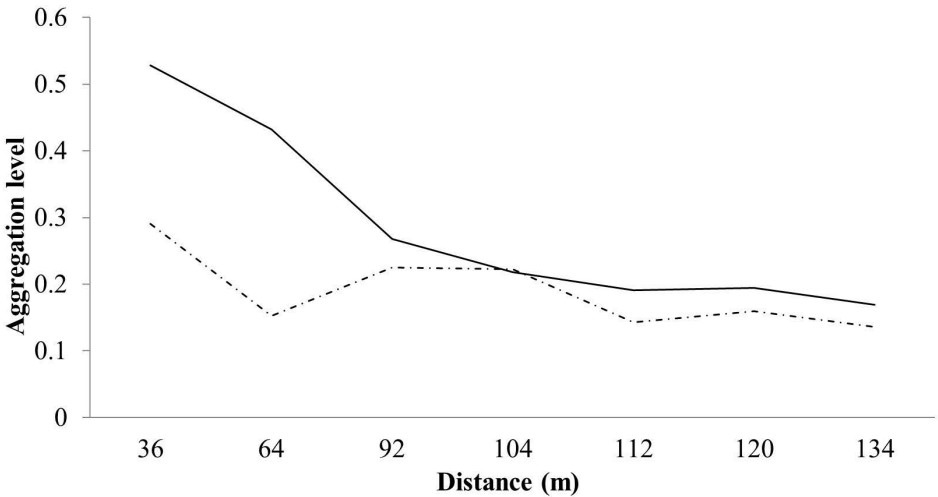

**Fig 5. Moran's I correlogram illustrating the aggregated spatial pattern among individuals with flowering (---) and fruiting (—) events in the first distance classes.**

**Table 2. Significant Spearman correlation ($r_s$) between the fortnightly means of climatic variables and the vegetative and reproductive phenological events of *Sarcomphalus joazeiro* Mart.**

| Variables/fortnights * | | Defoliation | Leaf budding | Flower buds | Flowers anthesis | Immature fruits | Ripe fruits |
|---|---|---|---|---|---|---|---|
| Air temperature (°C) | $r_{s1}$ | -0.552 | ns | 0.175 | 0.544 | 0.408 | ns |
| | $r_{s2}$ | -0.592 | ns | 0.374 | 0.514 | 0.501 | ns |
| | $r_{s3}$ | -0.547 | ns | 0.329 | 0.436 | 0.550 | 0.289 |
| | $r_{s4}$ | -0.510 | ns | 0.175 | 0.282 | 0.571 | 0.401 |
| Relative humidity (%) | $r_{s1}$ | ns | ns | -0.514 | -0.498 | ns | 0.452 |
| | $r_{s2}$ | 0.286 | ns | -0.553 | -0.554 | ns | 0.346 |
| | $r_{s3}$ | 0.496 | ns | -0.647 | -0.641 | ns | 0.022 |
| | $r_{s4}$ | 0.535 | ns | -0.553 | -0.603 | -0.357 | ns |
| Precipitation (mm) | $r_{s1}$ | ns | ns | -0.279 | ns | ns | ns |
| | $r_{s2}$ | ns | ns | -0.365 | -0.277 | ns | ns |
| | $r_{s3}$ | ns | ns | -0.365 | -0.335 | ns | ns |
| | $r_{s4}$ | 0.284 | ns | -0.346 | -0.348 | ns | ns |

*Correlations of the first (rS1), second (rS2), third (rS3) and forth (rS4) fortnights prior to the phenological event. ns: correlation not significant ($p > 0.05$).

## Biometry of fruits and seeds

The biometric data for *S. joazeiro* fruits were 2.73 g, 16.94 mm, 14.52 mm, and 16.34 mm for fresh weight, diameter, length, and thickness, respectively. Meanwhile, the average data for seeds were 0.41 g, 7.55 mm, 11.00 mm, and 7.11 mm for the characteristics mentioned (Table 3).

## Discussion

Phenological studies of species in the seasonal dry tropical forest (SDTF) are important to understand the patterns of flowering, fruiting, and other phases of the plant life cycle due to the marked variations between dry and rainy periods. However, the response of trees to

**Table 3. Means of biometric characteristics of fruits and seeds of *Sarcomphalus joazeiro*. N: sample size, CV: coefficient of variation, standard deviation, and mean.**

| Biometric characteristics | | N | Maximum | Minimum | Mean ± standard error | Standard deviation | CV (%) |
|---|---|---|---|---|---|---|---|
| Fruit | mass (g) | 200 | 4.25 | 1.30 | 2.73 ± 0.04 | 0.63 | 23.32 |
| | diameter (mm) | 200 | 20.10 | 12.10 | 16.94 ± 0.12 | 1.64 | 9.69 |
| | length (mm) | 200 | 17.80 | 11.60 | 14.52 ± 0.08 | 1.23 | 8.46 |
| | thickness (mm) | 200 | 19.70 | 12.00 | 16.34 ± 0.12 | 1.68 | 10.27 |
| Seed | mass (g) | 200 | 0.20 | 0.08 | 0.41 ± 0.01 | 0.10 | 24.90 |
| | diameter (mm) | 200 | 9.40 | 5.50 | 7.55 ± 0.05 | 0.76 | 10.17 |
| | length (mm) | 200 | 13.10 | 8.00 | 11.00 ± 0.05 | 0.80 | 7.13 |
| | thickness (mm) | 200 | 9.00 | 5.00 | 7.11 ± 0.05 | 0.70 | 9.74 |

drought in SDTF is complex. Species exhibit a range of strategies concerning leaf production and shedding, spanning the spectrum from deciduous to evergreen [38], which can affect the functional diversity of leaf traits due to aridity [39].

In some cases, underground water reserves and physiological adaptations mitigate the effects of rainfall scarcity, making phenological patterns weakly linked to environmental variables. This phenomenon is reflected in the vegetative phases of *S. joazeiro*. The species is considered evergreen with seasonal renewal [40], given that it produces new leaves before or concurrently with the loss of old leaves. Thus, despite a significant percentage of leaf area loss in the canopy, the species never becomes completely leafless, as leaf budding occurs simultaneously with the shedding of old leaves. This leaf exchange and flowering during the drought in *S. joazeiro* result from a series of mechanisms: efficient stomatal performance, coupled with the composition of its leaf epicuticular wax, characterized by a chemical constitution highly effective against water permeability [41]. Additionally, the extensive root system with deep radial roots and resilient hydraulic architecture (smaller vessel diameters and stronger vessel walls) can tolerate significant soil moisture limitations [42].

In this sense, water availability in dry forests significantly influences the development of certain plant species more than the precipitation volume. Some studies demonstrate the high drought tolerance of *S. joazeiro* [43]. These results contribute to the explanation by[44], who state that species with mechanisms for more efficient water acquisition or storage may exhibit phenological patterns independent of rainfall rates.

Thus, it was demonstrated that there is a low or no significant correlation of the species' phenology with precipitation, temperature, and relative air humidity, unlike the species *Geoffrea spinosa*, which, in a phenological study in the same fragment, showed correlations of meteorological variables with some phenophases [45]. This pattern suggests that it may be necessary to evaluate the species' interaction with the abiotic environment differently and reinforces how environmental behavior varies according to the studied species and its ecological mechanisms.

The *S. joazeiro* flower and fruit production are annual, with the synchrony and the flowering period observed as reproductive strategies. Populations that exhibit synchrony during flowering may attract more pollinators and enhance pollen flow, leading to increased fruit formation that contributes to the ecological success of the species [46,47]. The intensification of flowering between the classes of 92 to 104 meters may indicate that the species invested in a high intensity of flowers to achieve more significant pollination. However, the spatial distance between plants may have been less efficient in attracting vectors, as there was a decrease in the fruiting rate in plants that were spatially more distant.

In the case of this population of *S. joazeiro*, it can be asserted that the formation of flowers and, consequently, fruits occur in clusters. This is an important characteristic that contributes

to its reproductive strategy, as the species enhances the floral display, and combined with osmophores and pigments that reflect ultraviolet rays, they constitute an attractive ensemble for pollinators [48].

The fruiting period in the fragment coincides with what is observed in other regions, with ripe fruits available for collection in the first half of the year (January to July) [49]. This information is crucial for guiding the seed collection period for seedling production, which can be implemented for various purposes. However, the production of ripe fruits practically every month in the second year likely occurred due to disruptions stemming from anthropogenic pressures in the area, which may have affected maturation synchrony.

Regarding the interaction with fauna, [50], studying phenological data of 10 tree species in the Caatinga, observed that *S. joazeiro* is one of the few species that flowers during the dry season, potentially serving as an essential food resource for fauna during the drought. Its flowers are primarily visited by Hymenoptera from the families Apidae, Vespidae, Crabronidae, Leucospidae, and by Diptera from the families Syrphidae, Muscidae, Calliphoridae, Tabanidae, Dolichopodidae, Stratiomydae, Otitidae, and Micropezidae, with nectar being the main floral resource sought by the majority of visitors [51–53].

*Sarcomphalus joazeiro* exhibits a syndrome of zoochorous dispersal; its fruits ripen and become available for consumption during the rainy season, reaching the average occurrence date in April. Although it is a species highly tolerant to the lack of rain, precipitation can also favor fruit production [26]. The highest number of frugivorous birds are present in the Caatinga during increased rainfall, with *S. joazeiro* serving as a food source. The mammals *Mazama gouazoubira* and *Sapajus libidinosus* have been identified as potential dispersers [26].

Studying the biometrics of fruits and seeds from different *S. joazeiro* matrices, [54] found similar values to ours, observing fruits with an average length of 13.8 cm and a diameter of 18.6 cm, while seeds had a length of 10.6 cm and a diameter of 6.9 cm. Biometric analysis provides valuable information beyond the mere characterization of the species. These data are essential for informing conservation efforts, distinguishing between species, exploiting economic resources, assessing genetic variability among populations, and guiding genetic improvement. Additionally, analyzing pulp yield from fruits is crucial for both fresh consumption and processing purposes [55,56].

In this sense, it is emphasized that the protecting of various plant species is essential for future effective conservation strategies in the Caatinga, ensuring a constant presence of fruits [26]. These fruits provide propagation availability and the possibility of individual entry into the areas. Moreover, fleshy fruits may be a food source for wildlife, which is even more crucial during drought and food scarcity [57].

## Conclusions

The study provided valuable insights into the ecological dynamics of *S. joazeiro* and the strategies it adopts to ensure its success in the Caatinga environment. The phenology of *S. joazeiro* is notable for its uniqueness, with floral production occurring during the dry season and fruiting at the beginning of the rainy season. Flower and fruit production is annual, with reproductive strategies including synchrony, aggregated spatial patterns, and the timing of flowering. In terms of harvesting, April and May were identified as the most suitable months for collecting fruits and seeds.

The species is characterized as perennial with seasonal leaf renewal, demonstrating remarkable drought resistance, as evidenced by the weak correlation between its phenological performance and meteorological variables. This suggests that its phenology is regulated by additional abiotic factors or internal ecological mechanisms.

Overall, the research significantly contributes to the understanding of the ecological processes of *S. joazeiro* offering an in-depth analysis of the dynamics of seasonally dry tropical forest. In this sense, the scientific knowledge not only contributes to understanding the ecosystem but also to the development of effective conservation and sustainable management strategies.

## Supporting information

**S1 File. The spreadsheet (XLS file) contains the dataset used in the study.** The file includes circular statistics data and the data used to generate the figures presented in the manuscript. (XLSX)

## Author contributions

**Conceptualization:** Fernanda Moura Fonseca Lucas, Kyvia Pontes Teixeira das Chagas, Ageu da Silva Monteiro Freire, Fábio de Almeida Vieira.

**Data curation:** Fernanda Moura Fonseca Lucas, Kyvia Pontes Teixeira das Chagas, Ageu da Silva Monteiro Freire.

**Formal analysis:** Fernanda Moura Fonseca Lucas, Kyvia Pontes Teixeira das Chagas, Ageu da Silva Monteiro Freire, Vivian Raquel Bezerra de Sousa, Fábio de Almeida Vieira.

**Funding acquisition:** Fábio de Almeida Vieira.

**Investigation:** Fernanda Moura Fonseca Lucas, Kyvia Pontes Teixeira das Chagas, Ageu da Silva Monteiro Freire, Vivian Raquel Bezerra de Sousa, Fábio de Almeida Vieira.

**Methodology:** Fernanda Moura Fonseca Lucas, Kyvia Pontes Teixeira das Chagas, Ageu da Silva Monteiro Freire.

**Project administration:** Kyvia Pontes Teixeira das Chagas, Fábio de Almeida Vieira.

**Resources:** Ageu da Silva Monteiro Freire, Fábio de Almeida Vieira.

**Software:** Fernanda Moura Fonseca Lucas, Kyvia Pontes Teixeira das Chagas, Ageu da Silva Monteiro Freire, Vivian Raquel Bezerra de Sousa.

**Supervision:** Fábio de Almeida Vieira.

**Validation:** Fernanda Moura Fonseca Lucas, Kyvia Pontes Teixeira das Chagas, Ageu da Silva Monteiro Freire, Vivian Raquel Bezerra de Sousa, Fábio de Almeida Vieira.

**Visualization:** Fernanda Moura Fonseca Lucas, Kyvia Pontes Teixeira das Chagas, Ageu da Silva Monteiro Freire, Vivian Raquel Bezerra de Sousa.

**Writing – original draft:** Fernanda Moura Fonseca Lucas, Kyvia Pontes Teixeira das Chagas, Ageu da Silva Monteiro Freire, Vivian Raquel Bezerra de Sousa, Fábio de Almeida Vieira.

**Writing – review & editing:** Fernanda Moura Fonseca Lucas, Kyvia Pontes Teixeira das Chagas, Ageu da Silva Monteiro Freire, Vivian Raquel Bezerra de Sousa, Fábio de Almeida Vieira.

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
