## [Decision Letter · Decision Letter 0]

24 Oct 2024

PONE-D-24-27053Functional strategies of Sarcomphalus joazeiro in a Brazilian Seasonally Dry Tropical Forest FragmentPLOS ONE

Dear Dr. de Almeida Vieira,

Thank you for submitting your manuscript to PLOS ONE. After careful consideration, we feel that it has merit but does not fully meet PLOS ONE’s publication criteria as it currently stands. Therefore, we invite you to submit a revised version of the manuscript that addresses the points raised during the review process. The overall manuscript need to be revised. The reviewers has commented on Methodology, Results and Discussion and Conclusion part extensively. Also, the Keywords, References should be thoroughly revised as per the comments. The manuscript lack its flow in sentences, therefore, it is advised to check completely. The reference style should be unique overall. In present form, manuscript is not suitable for publication, and after making necessary changes, the manuscript may be considered for publication. 

We look forward to receiving your revised manuscript.

Kind regards,

Salman Khan, Ph.D.

Academic Editor

PLOS ONE

“This work was financed in part by the Office to Coordinate Improvement of Higher Education Personnel - Brazil (CAPES) - Finance Code 001. The authors acknowledge the Conselho Nacional de Desenvolvimento Científico e Tecnológico (CNPq) for their financial support (Grant no. 407700/2023-4).”

5. We note that Figure 1 in your submission contain copyrighted images. All PLOS content is published under the Creative Commons Attribution License (CC BY 4.0), which means that the manuscript, images, and Supporting Information files will be freely available online, and any third party is permitted to access, download, copy, distribute, and use these materials in any way, even commercially, with proper attribution. For more information, see our copyright guidelines: http://journals.plos.org/plosone/s/licenses-and-copyright.

6. We note that Figure 1 in your submission contain [map/satellite] images which may be copyrighted. All PLOS content is published under the Creative Commons Attribution License (CC BY 4.0), which means that the manuscript, images, and Supporting Information files will be freely available online, and any third party is permitted to access, download, copy, distribute, and use these materials in any way, even commercially, with proper attribution. For these reasons, we cannot publish previously copyrighted maps or satellite images created using proprietary data, such as Google software (Google Maps, Street View, and Earth). For more information, see our copyright guidelines: http://journals.plos.org/plosone/s/licenses-and-copyright.

1. You may seek permission from the original copyright holder of Figure(s) [#] to publish the content specifically under the CC BY 4.0 license. 

Reviewers' comments:

Reviewer's Responses to Questions

**Comments to the Author**

1. Is the manuscript technically sound, and do the data support the conclusions?

Reviewer #1: Partly

Reviewer #2: Yes

Reviewer #3: Yes

2. Has the statistical analysis been performed appropriately and rigorously? 

Reviewer #1: Yes

Reviewer #2: Yes

Reviewer #3: I Don't Know

3. Have the authors made all data underlying the findings in their manuscript fully available?

Reviewer #1: Yes

Reviewer #2: Yes

Reviewer #3: Yes

4. Is the manuscript presented in an intelligible fashion and written in standard English?

Reviewer #1: Yes

Reviewer #2: Yes

Reviewer #3: Yes

5. Review Comments to the Author

Reviewer #1: 1. The article (PONE-D-24-27053) mainly pertains to Functional strategies of Sarcomphalus joazeiro in a Brazilian Seasonally Dry Tropical Forest Fragment. The study is quite routine and not has much novelty for international readers. Besides, it can consider for its publication after resolving the issues or incorporating the suggestions. I had observed some major issues related to the MS. They are as under:

2. MS title is not perfectly suitable as per the study carried out. Must be change the title reflecting the centre theme or idea of the study.

3. Abstract is verbose and lacks clear delineation of objectives, significant finding of the findings. Include a clear mention of the research gap that your study addresses.

4. Authors didn’t focus much into need of phenological studies which is the lacuna in the Introduction part. Kindly provide more detailed information on the previous studies to better contextualize your research. First hypothesis is already an established fact then why did the authors choose this hypothesis

5. In methodology section, describe the method of selecting plants and sampling more rigorously to ensure representativeness and minimize bias. Authors didn’t include the pollen and stigma characteristics which are critical observations affected with meteorological parameters and play a vital role in climate change studies.

6. Results and discussion are well presented and comparing the observation with the previous literatures.

7. Some paragraphs in discussion part (line No 335 to 349) are deviating from the main focus and parameters. May omit these paragraphs.

8. In conclusion section, add some salient finding and don’t emphasize more on the future perspectives as it is research paper rather than review paper.

9. The main focus of manuscript is as per the journal’s name and criteria. Thus, I found the paper may be found suitable for the consideration of its publication in “PLOS ONE” journal and Requires Major Revision.

Reviewer #2: General comments

Abstracts: Abstract was not written well, thus try to refine it by adding the methodology and recommendation parts.

Key words: Capitalize the first letter of each word and add one more key word.

Statement of the problem: is not well stated; you stated that “to ensure success amidst a drought” but, the tree is already grown and growing in the area. Please write the main reason to study the functional strategies of S. joazeiro in a Brazilian Seasonally Dry Tropical Forest Fragment.

Methodology: the researchers used the presence of reproductive events and good phytosanitary conditions as tree selection criteria; but what about the effect of tree’s age on the phenology?

When you write the methodology of research article please, don’t personalize it. For example don’t use “we, I”

Result and discussion: written well

Reference: The reference writing style is not consistent. The researchers used et al in some parts of bibliography and not in others. I recommend writing the name of whole authors in bibliography. But it depends on the type of reference writing styles (Harvard, APA, MLA and others)

Reviewer #3: This manuscript highlights the seasonal drought adaptation strategies for a tree species Sarcomphalus joazeiro (Mart.). The tree species is ecologically integral for the maintenance of Flora and Fauna of Brazilian tropical forests. This study analysed the vegetative and reproductive growth of the tree S. joazeiro for two years and identified that the tree phenophases does not significantly correlates with the meteorological variables. Given the importance of the tree species in a forest ecology and understanding the tree physiology for better environment adaptability and forest conservation could be of significance importance. However, the important parameters to highlight the drought induction and tree response is not included in this study; for instance, relative leaf water content comparison during the rainfall (April) and relative drought season (October) for the forest should be supplemented. Along with the drought inducible enzymes and antioxidant activity should be added such as proline contents, SOD, POD, CAT, etc. to highlight either the trees are actually facing a drought condition or not. Which will further support the conclusion of current study.

The manuscript should be supplemented with the relative parameters prior to its acceptance in the Plos One Journal.

Minor mistakes:

Lines 210, 321, Table 2 (Defoliantion), Table 3 (lenght).

6. PLOS authors have the option to publish the peer review history of their article (what does this mean? ). If published, this will include your full peer review and any attached files.

**Do you want your identity to be public for this peer review?** For information about this choice, including consent withdrawal, please see our Privacy Policy .

Reviewer #1: **Yes: ** Dr. Ashok Kumar Dhakad, Scientist (Forestry), Department of Forestry & Natural Resources, Punjab Agricultural University, INDIA

Reviewer #2: No

Reviewer #3: No

---

## [Author Response · Author response to Decision Letter 1]

12 Dec 2024

JOURNAL REQUIREMENTS

Response: We have reviewed and ensured that our manuscript now meets PLOS ONE's style requirements, including those for file naming.

Response: Since it is an area belonging to the university, no authorization was necessary, with this information being placed in the manuscript "The study area is part of the campus of the Federal University of Rio Grande do Norte, and no authorization was required for access."

3. Thank you for stating the following financial disclosure: “This work was financed…” Please state what role the funders took in the study. If the funders had no role, please state: "The funders had no role in study design, data collection and analysis, decision to publish, or preparation of the manuscript."

Response: The requested information has been included in the manuscript.

Response: As suggested, we are sending the dataset in a spreadsheet, which is necessary to replicate the results.

5. We note that Figure 1 in your submission contain copyrighted images.

Response: We would like to clarify that Figures 1 (D-I) in our submission does not contain any copyrighted images. The photos included in this figure are the original work of the authors. To address this matter, we have included the following message in the figure caption: “The images (D-I), provided by the authors, are suitable for publication under the Creative Commons Attribution License (CC BY 4.0)”.

6. We note that Figure 1 in your submission contain [map/satellite]

Response: As requested, a replacement figure has been provided, which complies with the CC BY 4.0 license. The new location figure uses OpenStreetMap as the base map, which provides open data licensed under the Open Data Commons Open Database License (ODbL), managed by the OpenStreetMap Foundation (OSMF). To meet the requirement for credit attribution in the caption, I entered the following format, which adheres to the license terms: “…map data from OpenStreetMaps (accessed on 2024-12-05, openstreetmap.org/copyright)”.

REVIEWER 1

1. The article (PONE-D-24-27053) mainly pertains to Functional strategies of Sarcomphalus joazeiro in a Brazilian Seasonally Dry Tropical Forest Fragment. The study is quite routine and not has much novelty for international readers. Besides, it can consider for its publication after resolving the issues or incorporating the suggestions. I had observed some major issues related to the MS.

Response: Thank you. We have carefully reviewed the comments and suggestions provided by the reviewers and have made the necessary punctual corrections directly in the manuscript.

2. MS title is not perfectly suitable as per the study carried out. Must be change the title reflecting the centre theme or idea of the study.

Response: We appreciate the suggestion and have made the necessary adjustments. The new title is Phenological strategies of an evergreen tree in the Caatinga.

3. Abstract is verbose and lacks clear delineation of objectives, significant finding of the findings. Include a clear mention of the research gap that your study addresses.

Response: The abstract has been adjusted to improve reader understanding.

4. Authors didn’t focus much into need of phenological studies which is the lacuna in the Introduction part. Kindly provide more detailed information on the previous studies to better contextualize your research. First hypothesis is already an established fact then why did the authors choose this hypothesis.

Response: We have revised the Introduction to address the need for phenological studies and to better contextualize the research. The updated section now includes the following: “However, despite their importance, few studies provide ecological information on the phenology of forest species, especially in the Caatinga. Some more recent studies have carried out phenological analyses using satellite images, with the aim of optimizing field monitoring (Medeiros et al., 2022).” we reformulated the hypothesis to ensure its clarity.

5. In methodology section, describe the method of selecting plants and sampling more rigorously to ensure representativeness and minimize bias. Authors didn’t include the pollen and stigma characteristics which are critical observations affected with meteorological parameters and play a vital role in climate change studies.

Response: Thank you for your suggestion. We have included a more detailed description of the plant selection and sampling method in the “2.3. Phenological Data Collection” section, ensuring representativeness and minimizing bias. These revisions have been highlighted in red for clarity.

6. Results and discussion are well presented and comparing the observation with the previous literatures.

Response: Thank you for your positive feedback.

7. Some paragraphs in discussion part (line No 335 to 349) are deviating from the main focus and parameters. May omit these paragraphs.

Response: We appreciate the suggestion, the adjustments were made to make the text more coherent.

8. In conclusion section, add some salient finding and don’t emphasize more on the future perspectives as it is research paper rather than review paper.

Response: In line with the suggestions, the part emphasizing future perspectives was removed, leaving the text with emphasis on the results obtained.

9. The main focus of manuscript is as per the journal’s name and criteria. Thus, I found the paper may be found suitable for the consideration of its publication in “PLOS ONE” journal and Requires Major Revision.

Response: Thank you. We carefully addressed the points raised by all three reviewers, making the necessary corrections and revisions as indicated.

REVIEWERS COMMENTS IN .DOC ARCHIVE:

How the present study will help in conservation of these species?

Response: This study will contribute to the conservation of Sarcomphalus joazeiro in several ways. First, by understanding the phenological strategies of the species, such as identifying the low correlation between the phenophases of S. joazeiro and meteorological variables, we can highlight the species' adaptability to climate change. This suggests that it may be less vulnerable to extreme variations, making it a model for conservation efforts. Additionally, the analysis of reproductive synchrony and the biometrics of fruits and seeds provides valuable information for species management practices, such as the optimal timing for seed collection and seedling production. This is crucial for restoration programs.

It is an established fact that phenological events are strongly correlated or affected with the meteorological variables. then why the authors choose this hypotheses in the present study?

Response: We appreciate the comment and understand that the relationship between phenological events and meteorological variables is well-documented in the literature. However, the choice of this hypothesis for the present study is based on a specific phenological characteristic of Sarcomphalus joazeiro, commonly known as an evergreen tree that rarely or never sheds its leaves, even during drought periods. It is recognized that S. joazeiro is an extremely drought- and heat-adapted species. Therefore, through this hypothesis, we aim to investigate whether the leaf persistence of this species is indeed related to meteorological factors or if it possesses other mechanisms that sustain this behavior independent of weather conditions.

Authors are advised to include the pollen and stigma characteristics as they are very critical observations whose are eventually affected with the meteorological parameters.

Response: We appreciate your observation and agree that the characteristics of pollen and stigma are critical variables, with potential influence from meteorological factors. However, these aspects were not included in the present analysis. We will consider this recommendation in future studies to deepen our understanding of phenological interactions.

it is not acceptable thing that meteorological parameters did not affect the silvics of plant species.

Response: We agree with your statement. In this study, we also observed the existence of correlations between the phenological and meteorological variables analyzed. However, these correlations were classified as weak (r < 0.6). Although the evaluated meteorological parameters influence vegetative and reproductive phenology, their impact was limited. These results suggest that other factors not included in this study, such as soil water availability, may play a more significant role in the vegetative and reproductive development of the species. Therefore, the findings presented in this paper open new perspectives for future investigations on the phenological adaptations of S. joazeiro.

No novelty in this statement.

Response: Thank you. We have removed the mentioned statement.

Fruit and seed dimensions of different species are taxonomically different for all species. This is nothing new in these lines.

Response: Thank you. We have removed the sections that referenced other species.

Try to add some salients findings of the study in conclusions part.

Response: Thank you. We have revised the conclusions section. As requested, important insights from the research have been added to the conclusion.

This is not part of this study.

Response: Thank you. We have excluded the mentioned section.

REVIEWER #2: GENERAL COMMENTS

Abstracts: Abstract was not written well, thus try to refine it by adding the methodology and recommendation parts.

Response: The abstract has been rewritten to include more methodological information and recommendations (highlighted in red). Thank you.

Key words: Capitalize the first letter of each word and add one more key word.

Response: The requests have been addressed.

Statement of the problem: is not well stated; you stated that “to ensure success amidst a drought” but, the tree is already grown and growing in the area. Please write the main reason to study the functional strategies of S. joazeiro in a Brazilian Seasonally Dry Tropical Forest Fragment.

Response: Although it is widely recognized that S. joazeiro has high adaptability to withstand long periods of drought, the fundamental question is: what makes this species so distinct in terms of ecological strategies? This study aims to explain traits, particularly reproductive ones, that enable the success of S. joazeiro in such an adverse environment. By understanding these strategies, we can gain a better understanding of how the species thrives in an environment characterized by extreme climatic fluctuations.

Methodology: the researchers used the presence of reproductive events and good phytosanitary conditions as tree selection criteria; but what about the effect of tree’s age on the phenology?

Response: The age of the trees was not considered as a criterion in this study, as the phenological potential was not compared among the analyzed individuals. It is important to highlight that the study was conducted in natural vegetation, not in a planted system, which limits the possibility of controlling variables such as tree age. Additionally, estimating the age of species in dry forests presents significant challenges, particularly due to the difficulty in visualizing and counting growth rings, which are often indistinct in these environmental conditions.

When you write the methodology of research article please, don’t personalize it. For example don’t use “we, I”

Response: Understood. The corrections have been made to the methodology.

Result and discussion: written well

Response: Thank you for your positive feedback.

Reference: The reference writing style is not consistent. The researchers used et al in some parts of bibliography and not in others. I recommend writing the name of whole authors in bibliography. But it depends on the type of reference writing styles (Harvard, APA, MLA and others)

Response: Thank you for this observation. The references have been standardized following the APA style.

REVIEWER 3:

This manuscript highlights the seasonal drought adaptation strategies for a tree species Sarcomphalus joazeiro (Mart.). The tree species is ecologically integral for the maintenance of Flora and Fauna of Brazilian tropical forests. This study analysed the vegetative and reproductive growth of the tree S. joazeiro for two years and identified that the tree phenophases does not significantly correlates with the meteorological variables. Given the importance of the tree species in a forest ecology and understanding the tree physiology for better environment adaptability and forest conservation could be of significance importance. However, the important parameters to highlight the drought induction and tree response is not included in this study; for instance, relative leaf water content comparison during the rainfall (April) and relative drought season (October) for the forest should be supplemented. Along with the drought inducible enzymes and antioxidant activity should be added such as proline contents, SOD, POD, CAT, etc. to highlight either the trees are actually facing a drought condition or not. Which will further support the conclusion of current study.

The manuscript should be supplemented with the relative parameters prior to its acceptance in the Plos One Journal.

Response: The suggestions mentioned are extremely important but were not part of the scope of this research, so they could not be analyzed together with the other variables. We appreciate the suggestion and intend to incorporate these analyses into future research.

Minor mistakes: Lines 210, 321, Table 2 (Defoliantion), Table 3 (lenght).

Response: Thank you. The words were identified and corrected.

---

## [Decision Letter · Decision Letter 1]

30 Dec 2024

Phenological strategies of an evergreen tree in the Caatinga

PONE-D-24-27053R1

Dear Dr. de Almeida Vieira,

We’re pleased to inform you that your manuscript has been judged scientifically suitable for publication and will be formally accepted for publication once it meets all outstanding technical requirements.

Kind regards,

Salman Khan, Ph.D.

Academic Editor

PLOS ONE

Additional Editor Comments (optional):

Reviewers' comments:

Reviewer's Responses to Questions

**Comments to the Author**

1. If the authors have adequately addressed your comments raised in a previous round of review and you feel that this manuscript is now acceptable for publication, you may indicate that here to bypass the “Comments to the Author” section, enter your conflict of interest statement in the “Confidential to Editor” section, and submit your "Accept" recommendation.

Reviewer #1: All comments have been addressed

Reviewer #2: All comments have been addressed

Reviewer #3: All comments have been addressed

2. Is the manuscript technically sound, and do the data support the conclusions?

Reviewer #1: Yes

Reviewer #2: Yes

Reviewer #3: Partly

3. Has the statistical analysis been performed appropriately and rigorously? 

Reviewer #1: Yes

Reviewer #2: Yes

Reviewer #3: Yes

4. Have the authors made all data underlying the findings in their manuscript fully available?

Reviewer #1: Yes

Reviewer #2: Yes

Reviewer #3: Yes

5. Is the manuscript presented in an intelligible fashion and written in standard English?

Reviewer #1: Yes

Reviewer #2: Yes

Reviewer #3: Yes

6. Review Comments to the Author

Reviewer #1: Incorporated all comments raised by the reviewers. They have prepared article well now and exhaustive resources used which is considered of high economic values.

Reviewer #2: I read the document very well. accordingly all of my previous comments have been addressed. congratulations!

Reviewer #3: No more comments from my side, the author has addressed most of the concerns raised by other peer reviewers.

7. PLOS authors have the option to publish the peer review history of their article (what does this mean? ). If published, this will include your full peer review and any attached files.

**Do you want your identity to be public for this peer review?** For information about this choice, including consent withdrawal, please see our Privacy Policy .

Reviewer #1: **Yes: ** Dr. Ashok Kumar Dhakad

Reviewer #2: No

Reviewer #3: No

---

## [Editor Report · Acceptance letter]

PONE-D-24-27053R1

PLOS ONE

Dear Dr. de Almeida Vieira,

I'm pleased to inform you that your manuscript has been deemed suitable for publication in PLOS ONE. Congratulations! Your manuscript is now being handed over to our production team.

Kind regards,

on behalf of

Dr. Salman Khan

Academic Editor

PLOS ONE